# Melatonin Improves Turbot Oocyte Meiotic Maturation and Antioxidant Capacity, Inhibits Apoptosis-Related Genes mRNAs In Vitro

**DOI:** 10.3390/antiox12071389

**Published:** 2023-07-06

**Authors:** Jiarong Zhang, Feixia Li, Xiaoyu Zhang, Ting Xie, Hongyu Qin, Junxian Lv, Yunhong Gao, Mingyue Li, Yuntao Gao, Yudong Jia

**Affiliations:** 1Yellow Sea Fisheries Research Institute, Chinese Academy of Fishery Sciences, Qingdao 266071, China; 2College of Fisheries and Life Science, Shanghai Ocean University, Shanghai 201306, China; 3Qingdao National Laboratory for Marine Science and Technology, Qingdao 266237, China

**Keywords:** *Scophthalmus maximus*, oocyte culture and maturation, melatonin, antioxidative capacity, apoptosis

## Abstract

High-quality eggs are essential for the sustainability of commercial aquaculture production. Melatonin is a potent candidate for regulating the growth and maturation of oocytes. Therefore, research on the effect of melatonin on marine fish oocytes in vitro has been conducted. The present study successfully established a culture system of turbot (*Scophthalmus maximus*) oocytes in vitro and investigated the effect of melatonin on oocyte meiotic maturation, antioxidant capacity, and the expression of apoptosis-related genes. The cultures showed that turbot *Scophthalmus maximus* late-vitellogenic denuded oocytes, with diameters of 0.5–0.7 mm, had a low spontaneous maturation rate and exhibited a sensitive response to 17α, 20β-dihydroxyprogesterone (DHP) treatment in vitro. Melatonin increased by four times the rate of oocyte germinal vesicle breakdown (GVBD) in a concentration- and time-dependent manner. The mRNA of melatonin receptor 1 (*mtnr1*) was significantly upregulated in the oocyte and follicle after treatment with melatonin (4.3 × 10^−9^ M) for 24 h in vitro, whereas melatonin receptor 2 (*mtnr2)* and melatonin receptor 3 (*mtnr3)* remained unchanged. In addition, melatonin significantly increased the activities of catalase, glutathione peroxidase, and superoxide dismutase, as well as the levels of glutathione, while decreasing the levels of malondialdehyde and reactive oxygen species (ROS) levels in turbot oocytes and follicles cultures in vitro. *p53*, *caspase3*, and *bax* mRNAs were significantly downregulated in oocytes and follicles, whereas *bcl2* mRNAs were significantly upregulated. In conclusion, the use of turbot late-vitellogenesis oocytes (0.5–0.7 mm) is suitable for establishing a culture system in vitro. Melatonin promotes oocyte meiotic maturation and antioxidative capacity and inhibits apoptosis via the p53-bax-bcl2 and caspase-dependent pathways, which have important potential to improve the maturation and quality of oocytes.

## 1. Introduction

The production of a large number of high-quality eggs is crucial for maintaining the development of the aquaculture industry [1]. Egg quality is influenced by various factors, such as nutrition, photoperiod, temperature, salinity, husbandry practices, and stress [2,3]. Egg formation is a complex dynamic process associated with oogenesis, oocyte maturation, and ovulation [4]. Thus, fully understanding the mechanisms underlying the processes of oocyte growth and development, and how these processes are coordinated by oocyte cultures in vitro, is essential for identifying the factors that affect the quality of eggs. In general, fish oocytes are arrested in meiotic prophase I during growth, resume meiosis (maturation) under gonadotropin stimulation, and are again arrested in meiotic metaphase II after ovulation [5,6]. Germinal vesicles breakdown (GVBD) is considered a common criterion for oocyte meiotic maturation in vertebrates, including mammals and teleosts [7,8]. In vitro oocyte cultures have been widely applied to observe GVBD and elucidate the molecular, physiological, and biochemical mechanisms underlying oocyte maturation in humans [9], pigs [10], rats [11], rabbits [12], and bovines [13]. GVBD has already been studied extensively in catfish [14,15], zebrafish (*Danio rerio*) and other teleost fishes [16]. However, the detailed mechanisms of oocyte development in fish are not fully understood due to the diversity of fish species. In addition, fish oocyte cultures in vitro have only been established in the carp [17], zebrafish *Danio rerio* [18], Japanese eel *Anguilla japonica* [19], loach *Misgurnus fossilis* [20], and perch *Anabas testudineus* [21]. However, the platform of oocyte cultures in vitro in economic marine fishes remains largely unexplored.

Melatonin (N-aceyl-5-methoxytryptamine) is an indole derivative secreted rhythmically from the pineal gland, and it plays a key role in regulating the circadian clock by high blood levels at night and low levels during the day. Melatonin is also synthesized by a wide variety of organs, and it modulates various physiological functions other than circadian rhythms [22]. In teleosts, melatonin was detected in the ovaries of the numerous fish species, including Indian major carp (*Catla catla*) [23], medaka (*Oryzias latipes*) [24], and zebrafish. Melatonin could regulate reproduction in seasonal breeding teleost by acting as a physiological signal of environmental light–dark cycles [25,26]. Additionally, it indirectly regulates oocyte maturation by affecting the synthesis/release of the hypothalamic gonadotropin-releasing hormone (GnRH), kisspeptins, or the gonadotropin-inhibitory hormone (GnIH) [27,28,29]. Melatonin can also directly regulate oocyte maturation by binding to specific melatonin receptors (Mtnr1, Mtrn2, Mtnr3), and G protein-coupled receptors in fish [30,31,32]. It also exerts antioxidant effects to reduce the large amount of peroxides produced during oocyte maturation and alleviate the damage that free radicals cause to tissues [33]. Thus, these findings suggest that melatonin has potentially positive effects on oocyte maturation and quality.

Turbot *Scophthalmus maximus* is an economically important flatfish species that is widely farmed in Europe and China [34]. Over the past 30 years of its development, the annual production of turbot in China has remained at approximately 60,000 tons, which accounts for approximately 80% of the world’s total output of aquaculture turbot [35]. The reproductive season of turbot in the natural environment is from April to August and can be extended throughout the year by regulating the ambient temperature and photoperiod in captivity [36]. However, turbot cannot naturally spawn eggs in captivity, i.e., they need to be hand-stripped from ripe fish and artificially fertilized [37]. Numerous studies of turbot reproduction have focused on sex control [38], environmental impact (salinity temperature and photoperiod) [39,40,41], exogenous drug stimulation [42,43] and HPG axis-related genes functions during the reproductive cycle [37,44,45,46,47]. These studies have laid a solid foundation for further exploring the reproductive physiology and endocrinology of turbot. However, the lack of an in vitro assay to observe its maturation is a major hurdle for the functional analysis of turbot oocyte development. Accordingly, the present study aims to develop a turbot oocyte culture system in vitro and use this assay to investigate the effects of melatonin on the meiotic maturation, antioxidant capacity, and apoptosis-related gene expression of turbot oocytes. These findings could provide a valuable platform for understanding the mechanism of the meiotic maturation of oocytes and developing novel strategies to improve turbot oocyte quality in captivity.

## 2. Materials and Methods

### 2.1. Animals and Experimental Conditions

Sexually mature female turbots were obtained from Tianyuan Aquatic Products Co., Ltd. (Yantai, China). The fish (30 females) were kept in round tanks (30 m^3^ in volume) and supplied with recirculating water at 12–13 °C in a photoperiod (16 h light: 8 h dark). Water salinity, dissolved oxygen, and pH were maintained at 30.0 ± 1.0‰, 7.0 ± 2.0 mg/L, and 7.9 ± 0.3, respectively. Ammonia nitrogen was lower than 0.1 mg/L. The fish were fed twice a day using commercial diet (Hai Do, Santong Bio-engineering Weifang Co., Ltd., Weifang, China). The commercial diet contained 9.28% moisture, 50.17% crude protein, and 14.52% crude lipid of dry matter.

### 2.2. Chemical and Reagents

DHP (17α, 20β-dihydroxyprogesterone) and melatonin were obtained from Sigma Chemicals (St. Louis, MO, USA). DHP and melatonin were dissolved in dimethyl sulphoxide (DMSO). The chemicals were then diluted with Ca^2+^/Mg^2+^-free medium, which was used for experimental treatment. The vehicle concentration in the medium was ≤0.1%.

### 2.3. Isolation and Incubation of Oocyte

In order to confirm the optimal stages of ovarian development in turbot for oocyte isolation, the fish were anesthetized using 100 mg/L tricaine methanesulfonate (MS-222, Sigma, St. Louis, MO, USA). Ovarian tissues were removed from eighteen fish and placed in Bouin’s solution for hematoxylin and eosin (HE) staining in order to identify the developmental stage of turbot oocytes. The stages of ovarian development were classified based on oocyte morphology [37]. Body and ovary weights were recorded in order to calculate the gonadosomatic index (GSI = gonad weight/body weight). The proportion of oocytes at different developmental stages was analyzed during ovarian development. The ovaries with a high proportion of late-vitellogenesis oocytes were used for in vitro cultures in the current study.

The turbot oocytes were separated using a mechanical method according to previous studies in zebrafish [48,49] and perch [21], with some modifications. Briefly, after fasting and anesthetization, the healthy follicles were separated from turbot late-vitellogenesis ovaries using forceps, and then washed in cold phosphate-buffered saline (Ca^2+^-Mg^2+^-free) solution with penicillin–streptomycin (Solarbio, Beijing China) three times. The follicles were separated into two groups. In one group, the follicles were digested with 0.3 mg/mL collagenase (Solarbio, Beijing, China) in a shaking water bath at 20 °C for 40 min in order to remove the follicular cell layer. After enough denuded oocytes were obtained, the oocytes were washed with fresh Ca^2+^-Mg^2+^-free DMEM medium at least three times to remove the remaining collagenase. In the other group, the follicles remained in fresh Ca^2+^-Mg^2+^-free DMEM medium and were left untreated with collagenase. Denuded oocytes and remaining follicles (untreated with collagenase) with three different diameters (<0.5 mm, 0.5–0.7 mm, >0.7 mm) were carefully selected and pooled. The same diameters of oocytes and follicles were randomly distributed in wells of a 24-well plate (Saiguo Biotech Co., LTD., Guangzhou, China) at a density of 50 oocytes/well or 50 follicles/well in 1 mL Ca^2+^/Mg^2+^-free DMEM medium at 14.5 °C. Germinal vesicle breakdown (GVBD) is an easily identifiable marker for oocyte maturation. The follicles and denuded oocytes were incubated for 24 h in the presence or absence of DHP (10^−6^ M) in order to score the GVBD and identify the optimal size of denuded oocytes to be used for in vitro studies.

To further identify the response of optimal size of denuded oocytes under oocyte-maturation-related hormone treatment, the GVBD rate of the oocytes was detected. The denuded oocytes (0.5–0.7 mm) and follicles were isolated and incubated with DHP (10^−6^, 10^−7^, 10^−8^, 10^−9^ M) and melatonin (4.3 × 10^−8^, 4.3 × 10^−9^, 4.3 × 10^−10^, 4.3 × 10^−11^ M) for 24 h. The control group received only the vehicle. The GVBD of denuded oocytes and follicles were counted. The specimens of the melatonin treated were collected and divided into two groups. One was snap-frozen in liquid nitrogen or preserved in RNAstore Reagent (Tiangen Biotech, Beijing, China) and then stored at −80 °C until RNA extraction, and the other was collected and then stored at −20 °C to analyze CAT, GSH-Px, and SOD activity as well as MDA, GSH, ROS, and cAMP content. All experiments were repeated three times to confirm the results. Triplicate wells were used for each endpoint or treatment within an experiment; therefore, a total of nine samples were collected per group in the current study.

### 2.4. Antioxidant Capacity Assay

Oocytes and follicles were homogenized in an ice bath with 0.9% normal saline (pH 7.2) with a ratio of 1:5 (*W*:*V*) and were then centrifuged at 4 °C for 10 min (7 × 103 g). The supernatant was collected immediately for the following analysis. SOD, CAT, GSH-Px activity, as well as the MDA, GSH, ROS, and cAMP content, was measured using corresponding commercial detection kits in accordance with the manufacturers’ instructions (Nanjing Jiancheng Bioengineering Institute, Nanjing, China). The SOD, CAT and GSH-Px activity were expressed as U/mg protein. MDA and GSH content were expressed as mmol/mg protein. The protein was determined using commercial bicinchoninic acid (BCA) detection kits in accordance with the manufacturers’ instructions (Nanjing Jiancheng Bioengineering Institute, Nanjing, China). cAMP was expressed as nmol/L. ROS were analyzed using fluorescent probe 2′,7′-dichlorofluorescin diacetate (DCFH-DA) and expressed as DCF fluorescence. The intra- and inter-assay coefficients of variation were less than 5%.

### 2.5. RNA Isolation, Reverse Transcription, and Quantitative Real-Time RT-PCR (qRT-PCR)

A two-step, real-time RT-PCR was performed to detect the expression of melatonin receptors (*mtnr1*, *mtnr2*, *mtnr3*) and apoptosis-related genes (*bax*, *bcl2*, *p53*, *caspase3*). Total RNA was extracted from the collected samples using SteadyPure Universal RNA Extraction Kit (Accurate Biotechnology, Changsha, Hunan, China) and was then quantified using a NanoDrop 2000 (Thermo Fisher Scientific, Rockford, IL, USA). The extracted total RNA was treated for 30 min with DNase I (Qiagen, Singapore) at 37 °C in order to prevent genomic DNA contamination. Subsequently, 1 μg of total RNA was reverse-transcribed using an Evo M-MLV RT Mix Kit (Accurate Biotechnology, Hunan, China) in accordance with the manufacturer’s instructions. The mRNA expression levels of *mtnr1*, *mtnr2*, *mtnr3*, *p53*, *bax*, *bcl2*, and *caspase3* were assessed via real-time RT-PCR using a SYBR Green Premix Pro Taq HS qPCR Kit II (Accurate Biotechnology, Hunan, China) and ABI-7500 Detection System (Applied Biosystems, Foster City, CA, USA). A 20 μL reaction volume for amplification contained 10 μL 2× SYBR Green Pro Taq HS Premix II, 0.8 μL of each primer (10 μM), 0.4 μL ROX Reference Dye (4 μM), 2 μL cDNA Template (25 ng/μL), and 6 μL sterile distilled water. Initial denaturation was performed at 95 °C for 30 s, followed by 40 cycles of amplification at 95 °C for 5 s and 60 °C for 30 s. The primer sequences and the PCR product lengths for *mtnr1*, *mtnr2*, *mtnr3*, *p53*, *bax*, *bcl2*, and *caspase3* are listed in Table 1. β-actin was selected as the housekeeping gene based on our previous study [50]. The expressions of genes were normalized to *β-actin* and expressed as folded change relative to the expression level in the control according to the 2^−∆∆CT^ method [51]. All samples were amplified in triplicates.

### 2.6. Statistical Analysis

Data are expressed as mean ± standard error of mean (SEM) of three independent experiments. All percentage data were subjected to arcsine square-root transformation before statistical analysis and then analyzed either via a Student’s *t*-test or one-way analysis of variance (ANOVA) followed by Bonferroni post hoc analysis using SPSS software, version 22.0 (SPSS, Inc. Chicago, IL, USA). A probability of *p* < 0.05 was considered to be statistically significant.

## 3. Results

### 3.1. Stage-Dependent Maturational Competence of Turbot Oocytes

Five stages were observed in the ovarian development of the female turbot based on oocyte morphology (Figure 1A). Stage III oocytes were characterized by an increasing accumulation of vitellogenic granules and the start of nucleic migration toward the cell periphery (Figure 1A). The gonadosomatic index (GSI) significantly increased from the previtellogenic (Prevtg) to the migratory nuclei (Mig-nucle) stage, with the highest values observed at the Mig-nucle stage, and then significantly decreased at the atresia (Atre) stage (Figure 1B, *p* < 0.05). However, no significant difference in GSI was found between the late-vitellogenic (Latvtg) and Mig-nucl stages (Figure 1B, *p* > 0.05). In addition, the proportion of stage III oocytes at the Latvtg stage was significantly higher than that of Mig-nucl during the development of ovaries (Figure 1C, *p* < 0.05). The germinal vesicle (GV) of immature stage III oocytes was placed centrally, and then oocytes turned translucent after maturation due to GVBD (Figure 2A). The denuded immature oocytes demonstrated a smooth edge after the removal of follicular cell layers (granulosa and theca cells) through collagenase treatment (Figure 2B). The spontaneous maturational competence of stage III oocytes varied with their diameters in in vitro cultures. The diameters of oocyte and follicles were mainly divided into three types (<0.5 mm, 0.5–0.7 mm, and >0.7 mm), of which more than 80% were 0.5–0.7 mm oocytes (Appendix A). The highest GVBD rate was observed in oocytes with diameters of more than 0.7 mm (Figure 2C, *p* < 0.05). Oocytes less than 0.7 mm in diameter had lower spontaneous maturational competence in vitro, whereas oocytes with 0.5–0.7 mm diameters exhibited a sensitive response to DHP treatment (Figure 2C, *p* < 0.05). Meanwhile, follicles that were 0.5–0.7 mm in diameter exhibited no response to DHP treatment (Figure 2C, *p* > 0.05). Therefore, late-vitellogenic oocytes with diameters of 0.5–0.7 mm were used for all subsequent experiments.

### 3.2. Effects of Melatonin on Meiotic Maturation of Turbot Oocytes

Similar to DHP, melatonin significantly increased the GVBD rate of the oocytes in a time- and dose-dependent manner (Figure 3A,B). The GVBD of the oocytes was initiated after a 6 h treatment with melatonin (4.3 × 10^−8^, 4.3 × 10^−9^, and 4.3 × 10^−10^ M) and DHP (10^−6^, 10^−7^, and 10^−8^ M). The GVBD rate significantly increased after melatonin (4.3 × 10^−8^, 4.3 × 10^−9^, and 4.3 × 10^−10^ M) and DHP (10^−6^, 10^−7^, and 10^−8^ M) treatments for 24 h compared with the control. Melatonin significantly stimulates the GVBD rate of turbot oocytes by treating the follicles from 4.3 × 10^−8^ to 4.3 × 10^−10^ M (Figure 3C, *p* < 0.05). The GVBD rate demonstrated no significant differences after the treatment of follicles with DHP from 10^−6^ to 10^−9^ M (Figure 3D, *p* > 0.05). The survival rate of oocytes and follicles cultured in vitro reached more than 95% for the control, DHP and the melatonin treatment group (Appendix A).

### 3.3. Effects of Melatonin on Mtnrs Expression and Antioxidant Capacity in Turbot Oocytes

*mtnr1*, *mtnr2*, and *mtnr3* were all detected in follicles and oocytes, while *mtnr1* had the highest values (Figure 4A, *p* < 0.05). The mRNA levels of *mtnr1* in follicles are significantly higher than those in oocytes, whereas *mtnr2* and *mtnr3* in follicles were significantly lower than those in oocytes (Figure 4A, *p* < 0.05). Melatonin (10^−6^ g/mL) significantly increased the mRNA expression level of *mtnr1* in the follicles (Figure 4B, *p* < 0.05), but *mtnr2* and *mtnr3* remained unchanged. Meanwhile, melatonin did not significantly affect the mRNA expression levels of *mtnr1*, *mtnr2*, and *mtnr3* in turbot oocytes in vitro (Figure 4C, *p* > 0.05).

Interestingly, melatonin significantly stimulated CAT, GSH-Px, and SOD activities and as well as GSH content in turbot oocytes and follicles after a 24 h treatment (Figure 5A–D, *p* < 0.05). The MDA and ROS levels in turbot oocytes and follicles significantly decreased subject to melatonin treatment (Figure 5E,F, *p* < 0.05).

### 3.4. Effects of Melatonin on Apoptosis-Related Gene Expression

The mRNA levels of the main genes involved in cell apoptotic signaling processes in the oocytes and follicles of turbot were altered under 24 h melatonin treatment (Figure 6). The mRNA levels of *p53*, *caspase3*, *bax,* and *bax*/*bcl2* significantly decreased under melatonin treatment for 24 h (Figure 6A–C, *p* < 0.05). Melatonin significantly upregulated *bcl2* mRNAs (Figure 6D,E, *p* < 0.05).

## 4. Discussion

An egg is the final product of oocyte growth and differentiation. The developmental status of oocytes is closely associated with subsequent fertilization and embryonic development, thereby assessing the quality of oocytes before ovulation is of great practical benefit during fish farming. In general, oocyte growth is arrested at the diplotene stage of the first meiotic division prophase for a few days or months in the most teleosts, and this process is accompanied by yolk accumulation [52]. Meiotic resumption and the maturation of fish oocytes generally occur at or near the time of vitellogenic growth completion; therefore, late-vitellogenic oocytes are usually considered suitable for in vitro cultures [53,54]. Dynamic models of final oocyte maturation have been identified in various fish species, including Eurasian perch (*Perca fluviatilis)* [55] pikeperch (*Sander lucioperca)* [56], ruffe (*Gymnocephalus cernua)* [57] and Burbot (*Lota lota)* [58]. In the present study, the yolk granules almost filled the ooplasm in stage III turbot oocytes. Similar results were reported in zebrafish (*Danio rerio)* [59] and European silver eel (*Anguilla Anguilla*) [60,61]. Meanwhile, turbot oocytes exhibit group-synchronous development throughout the reproductive season [34]. In the current study, the abundant stage III oocytes were mainly observed in the ovaries at the late-vitellogenesis stage, which was thus used for oocyte isolation and in vitro cultures. GVBD is an easily identifiable marker for oocyte maturation in vitro and correlated with oocyte size. Excessively large oocytes could result in a high GVBD rate culture in vitro. By contrast, small oocytes exhibit poor responses to external reproductive hormones. The screening of suitable oocytes by diameter has been conducted in zebrafish and Korean spotted sea bass (Lateolabrax maculatus) [16,48,62]. In the current study, late-vitellogenic denuded oocytes with diameters of 0.5–0.7mm had a low spontaneous maturation rate and exhibited sensitivity to DHP treatment. In addition, DHP can stimulate turbot oocyte GVBD in a time- and dose-dependent manner, similar to those in the chub mackerel (*Scomber japonicus)* [63], European sea bass (*Dicentrarchus labrax)* [64] and goldfish (*Carassius auratus)* [65,66]. These results suggest that turbot late-vitellogenic denuded oocytes (0.5–0.7 mm) are best used to establish a culture system in vitro. Therefore, the established turbot oocyte in vitro system may serve as a platform to investigate the meiotic maturation and the underlying mechanisms of oocytes under various internal and external factors.

Photoperiod is one of the main environmental factors that affects the seasonal reproduction of fish [65,66,67,68]. Melatonin is the key hormone that regulates photoperiod-induced physiological activity during the life cycle [22,69,70]. It can also affect the development of fish ovaries and modulate the maturation of oocytes by activating or inhibiting the secretion of HPG axis-related hormones [27,29]. In the present study, melatonin significantly stimulated the meiotic maturation of turbot oocytes in a time- and dose-dependent manner in vitro. Similar results were found in carp [71], zebrafish [72], and killfish (Fundulus heteroclitus) [73]. The direct actions of melatonin occur through binding to its specific receptors (Mtnr1, Mtnr2, and Mtnr3) [74]. Mtnr1 and Mtnr2 are found in all vertebrate species, whereas Mtnr3 is restricted to fish, amphibians, and birds [75]. Sakai et al. [32] also reported the fourth specific receptor in teleostean species and some reptiles, termed MTNR1a-like. Chattoraj et al. [76] found that the Mtnr1 protein has a greater expression in the membrane fraction than in the cytosol of the carp ovary homogenate. In the present study, *mtnr1* demonstrated higher mRNA levels than mtnr2 and *mtnr3* in follicles and denuded oocytes in turbot. Zhao et al. [77] found high mRNA levels of *mtnr1* and *mtnr2* during turbot ovarian development. The *mtnr1* mRNA is significantly upregulated in oocytes and follicles after melatonin treatment, whereas those of *mtnr2* and *mtnr3* remained unchanged in the current study. Melatonin could accelerate the action of the maturation-inducing hormone (MIH) on carp oocyte maturation in vitro in a melatonin-receptor-dependent manner [76]. It has been highlighted that *mtnr1* is expressed in maturing oocytes of goats and bovine animals, and that it promotes oocyte maturation by reducing cyclic adenosine monophosphate (cAMP) levels [78,79,80]. High intraoocyte cAMP levels can maintain the arrest of the first meiotic maturation of oocytes for a long time. The oocytes will resume meiosis and mature either when the synthesis of cAMP is downregulated or degraded [81]. Downregulated adenylyl cyclase 3 α (*ac3α*) and decreased cAMP content were observed in the oocytes and follicles of turbot after melatonin treatment (Appendix A). These results and the above literature suggest that *mtnr1* mediates the melatonin-induced meiotic maturation of turbot oocytes in vitro.

The physiological status of oocytes directly affects subsequent egg quality and correlates with the sustainability of commercial aquaculture production [3]. Excessive accumulation of free radicals in oocytes could induce oxidative stress, accelerate oocyte aging, and reduce egg quality [82,83]. Melatonin can directly enter cells and combine with some types of free radical molecules to exert antioxidant effects because of their highly lipophilic and hydrophobic properties [84,85,86]. Meanwhile, they also indirectly increase antioxidant enzyme activity and mRNA expression via their membrane receptors of melatonin [87]. In the present study, melatonin significantly stimulated SOD, CAT, and GSH-Px activities and GSH levels, as well as reducing MDA and ROS contents in turbot oocytes and follicles. In female reproduction, melatonin is associated with improving ovary function and oocyte quality by suppressing oxidative stress [87]. Melatonin enhanced the antioxidant capacity of cryopreserved ovaries via stimulating antioxidant enzymes in rats and mice [88,89]. Melatonin supplementation in vitro also significantly promotes oocyte maturation by increasing GSH levels and reducing ROS [79,84,90]. Improved egg quality was observed in mammals such as rats [83], pigs [91], and bovines [92,93] following a moderate supplementation of melatonin. In addition, melatonin can reduce MDA content and increase SOD, CAT, and GSH-Px activities in the follicular cells of Indian carp [23,92]. Therefore, melatonin can enhance the antioxidant activities of oocytes in vitro and potentially improve turbot egg quality.

Apoptosis is a highly regulated process of programmed cell death that plays a critical role in maintaining normal tissue homeostasis by eliminating dysfunctional cells [94]. P53, Bax, Bcl2, and caspase3 are important molecules in the apoptotic signaling pathway. P53 has been recognized for its critical function in initiating apoptosis by activating pro-apoptotic factor Bax, while suppressing antiapoptotic Bcl2 factors [95]. An essential feature of apoptosis is the release of cytochrome c from the mitochondria, which is regulated by the balance between pro-apoptotic and anti-apoptotic protein members of the Bcl2 family. Bax can induce apoptosis via the release of cytochrome c into the cytosol and activation of the caspase signaling pathway, whereas antiapoptotic protein Bcl2 inhibits the release of cytochrome c from the mitochondria [96]. Caspase3 is the main “effector caspase”, which executes the final steps of apoptosis in the extrinsic (death-receptor-mediated) and intrinsic (mitochondria-derived) apoptotic pathways [97]. Melatonin could inhibit endometrial epithelial apoptosis by downregulating *caspase 3* expression and the *bax*/*bcl2* ratio [98]. Xing et al. [99] reported that the supplementation of melatonin in vivo efficaciously suppressed mitochondrial dysfunction and the accompanying apoptosis by regulating *bcl2* and *bax* expression in mouse oocytes. In the current study, melatonin significantly upregulated *bcl2* and downregulated *p53*, *bax*, and *caspase3* mRNAs in oocytes and follicles of turbot. In addition, an increase in bax/bcl2 was also observed after the melatonin treatment. The ratio of Bcl2 to Bax reflects the anti-apoptosis capacity, further indicating the apoptosis levels [100]. An increase in bax/bcl2 ratio is a decisive factor that is positively correlated with apoptosis or cell death [101,102]. Crucially, melatonin can also ameliorate oxidative, stress-induced apoptosis in porcine oocytes [103]. Thus, these results and the aforementioned changes in the antioxidant capacity indicate that melatonin inhibited apoptosis via the p53-bax-bcl2 and caspase-dependent pathways and was beneficial for improving oocyte quality.

## 5. Conclusions

In summary, the current study is the first to establish a high-survival culture system of turbot oocytes in vitro and investigates the effect of melatonin on oocyte meiotic maturation. Late-vitellogenic oocytes with diameters of 0.5–0.7 mm, had a low spontaneous maturation rate and exhibited sensitivity to DHP treatments in a time- and dose-dependent manner, suggesting their suitability for establishing a culture system in vitro. Melatonin significantly stimulated the turbot oocyte GVBD rate and upregulated its specific receptor *mtnr1* mRNA. It also increased CAT, GSH-Px, and SOD activities, along with GSH contents, and it decreased the MDA and ROS levels of oocytes and follicles. Meanwhile, melatonin suppressed the pro-apoptotic genes (e.g., *p53*, *caspase3* and, *bax*), while activating antiapoptotic genes *bcl2*. Taken together, we successfully established a turbot oocyte culture system in vitro and confirmed that melatonin can stimulate oocyte meiotic maturation, promote the antioxidative capacity, and inhibited apoptosis via *p53*-*bax*-*bcl2* and caspase-dependent pathways. The specific receptor *mtnr1* was involved in the aforementioned action of melatonin, whereas its detailed mechanism needs to be further studied in the future. These findings provide a valuable platform for observing the dynamic process of final oocyte maturation and elucidating the underlying mechanisms of oocyte meiotic maturation, thereby developing a novel strategy for improving oocyte quality in turbots in captivity.

## Figures and Tables

**Figure 1 antioxidants-12-01389-f001:**
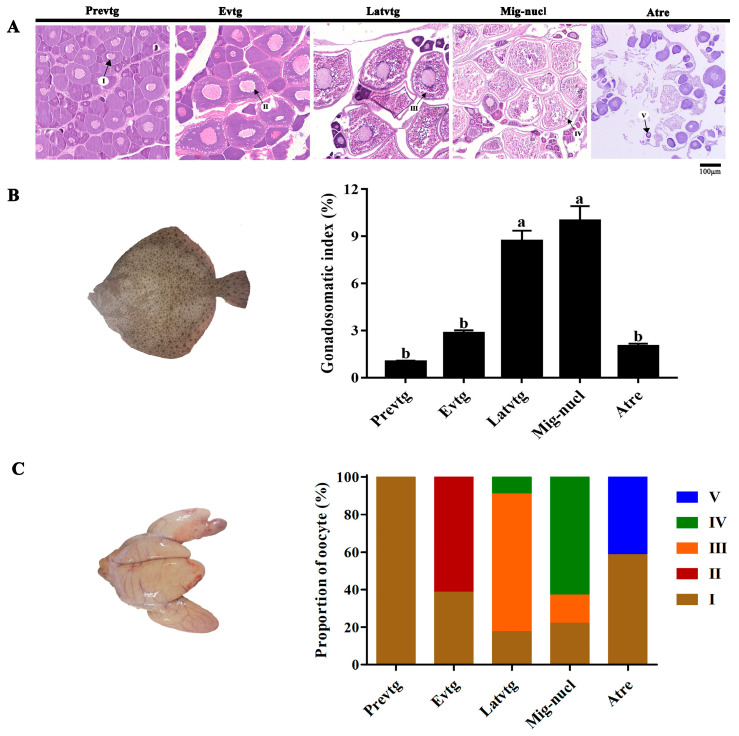
Ovarian development of turbot *Scophthalmus maximus* during reproductive cycle. (**A**) Development and types of turbot oocyte: I, arrows indicate oocytes at the stage of previtellogenesis and their nucleoli are on the periphery of the germinal vesicle. II, arrows indicate oocytes at the stage of early vitellogenesis and gradual accumulations of yolk granules in the central region. III, arrows indicate oocytes at the stage of late vitellogenesis, the yolk granules almost fill the ooplasm, and the nucleus has not yet begun to migrate peripherally. IV, arrows indicate oocytes at the stage of migratory nucleus, the yolk granules have attained their maximum size just prior to spawning, and the nucleus is not evident. V, arrows indicate oocytes at the stage of atresia have shrinkage or collapse. (**B**) Gonadosomatic index of female turbot during ovarian development. Values are represented as the mean ± SEM (*n* = 6). Bars with different superscript letters are significantly different (*p* < 0.05). (**C**) Proportions of oocytes at different developmental stages in turbot ovary. I, II, III, IV, V represent five different developmental stages of oocytes. Prevtg, previtellogenic; Evtg, early vitellogenic; Latvtg, late vitellogenic; Mig-nucle, migratory nuclei; Atre, atresia.

**Figure 2 antioxidants-12-01389-f002:**
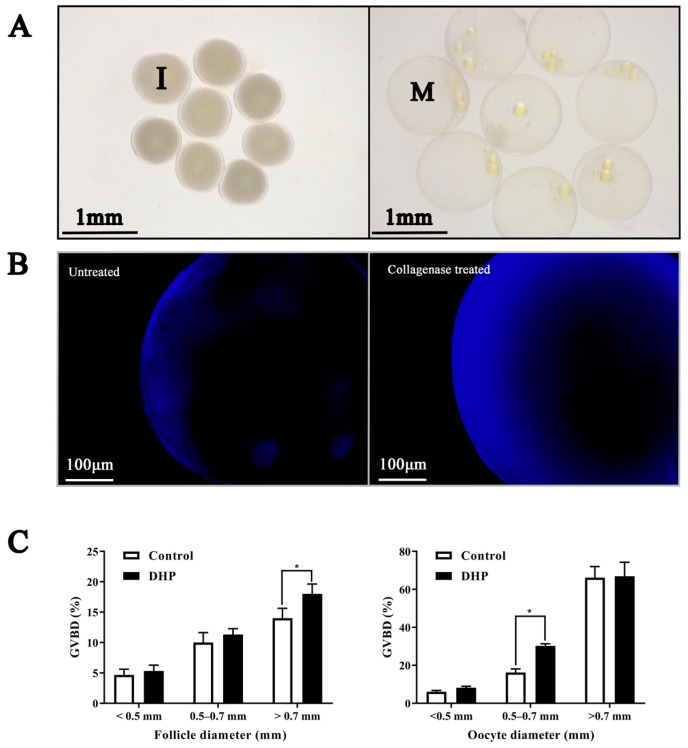
Isolation and culture of turbot *Scophthalmus maximus* oocytes in vitro. (**A**): Isolated late-vitellogenic oocytes of turbot. I: immature oocytes; M: maturation oocytes. (**B**) Removal of follicle cell layers by collagenase treatment. The left figure shows untreated follicle-enclosed oocytes and the right figure shows a collagenase-treated follicle; the denuded oocyte has a smooth edge without follicular cell layers. (**C**) Stage-dependent responsiveness of turbot follicles and oocytes to DHP stimulation (10^−6^ M). Values are represented as the mean ± SEM (*n* = 3). * Indicates significant difference (*p* < 0.05). DHP: 17α-Hydroxyprogesterone; GVBD: germinal vesicle breakdown.

**Figure 3 antioxidants-12-01389-f003:**
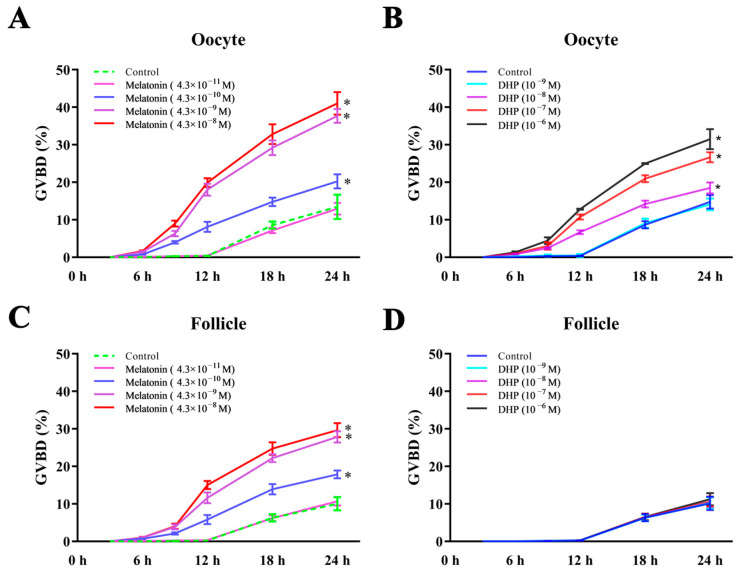
Time–dose response of the maturational competence of turbot (*Scophthalmus maximus)* oocytes induced by melatonin (**A**,**C**), and DHP (**B**,**D**). The GVBD rate was recorded at treatment for 6 h, 12 h, 18 h, 24 h. Each value represents the mean ± SEM of three replicates. An asterisk represents a significant difference (*p* < 0.05) compared with the control group. DHP: 17α, 20β-dihydroxyprogesterone; GVBD: germinal vesicle breakdown.

**Figure 4 antioxidants-12-01389-f004:**
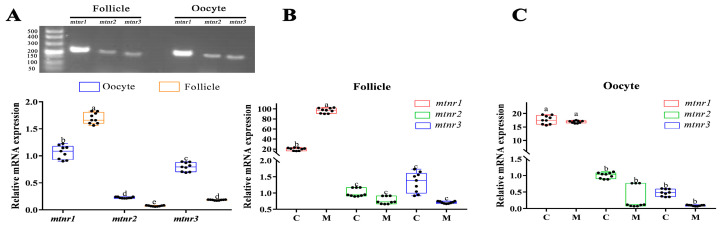
Effect of melatonin on its receptor expression in turbot *Scophthalmus maximus*. (**A**) Detection of melatonin receptors (*mtnr1*, *mtnr2, mtnr3*) expression in turbot follicle and oocyte (**B**,**C**) Expression of melatonin receptors (*mtnr1*, *mtnr2*, *mtnr3*) in turbot follicles and oocytes after treatment with melatonin (4.3 × 10^−9^ M) for 24 h. Values are represented as the mean ± SEM (*n* = 9). Bars with different superscript letters are significantly different (*p* < 0.05). *mtnr1*: melatonin receptor 1; *mtnr2*: melatonin receptor 2; *mtnr3*: melatonin receptor 3.

**Figure 5 antioxidants-12-01389-f005:**
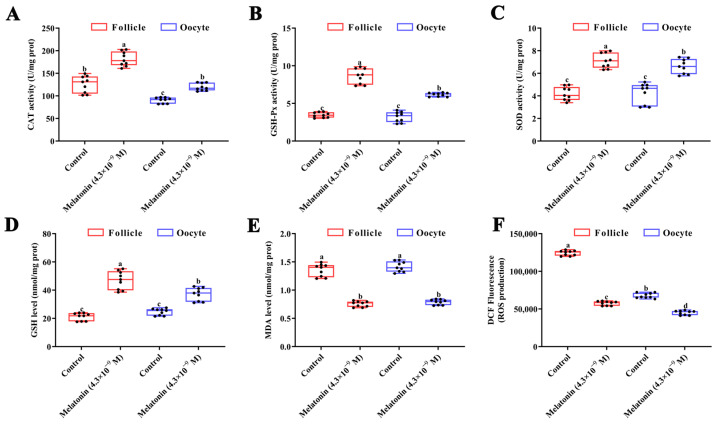
Effect of melatonin on antioxidant capacity in turbot (*Scophthalmus maximus)* follicles and oocytes after treatment for 24 h. (**A**) CAT activity; (**B**) GSH-Px activity; (**C**) SOD activity; (**D**) GSH content; (**E**) MDA levels; and (**F**) ROS levels. Values are represented as the mean ± SEM (*n* = 9). Bars with different superscript letters are statistically different (*p* < 0.05). CAT: catalase; GSH-Px: glutathione peroxidase; SOD: superoxide dismutase; GSH: glutathione; MDA: malondialdehyde; and ROS: reactive oxygen species.

**Figure 6 antioxidants-12-01389-f006:**
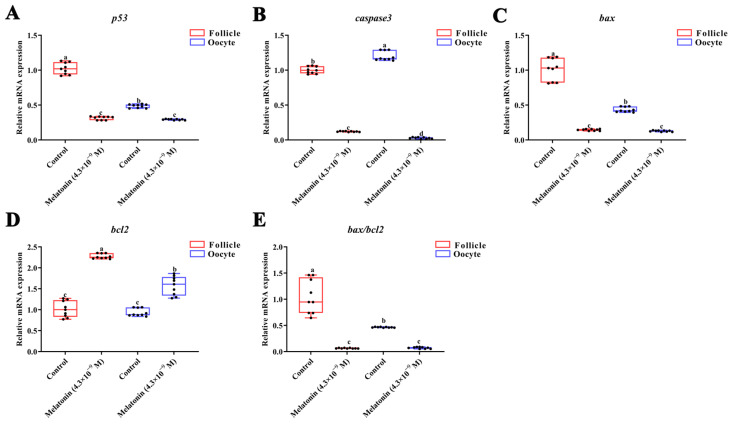
Expression of apoptosis-related genes after treatment with melatonin for 24 h. (**A**) *p53*, (**B**) *caspase3*, (**C**) *bax*, (**D**) *bcl2*, (**E**) *bax*/*bcl2* in turbot *(Scophthalmus maximus)* follicles and oocytes. Values are represented as the mean ± SEM (*n* = 9). Bars with different superscript letters are statistically different (*p* < 0.05).

**Table 1 antioxidants-12-01389-t001:** Primer sequences used in real-time RT-PCR.

Primers	Accession No.	Primer Sequence (5′ to 3′)	Product Length (bp)
Mtnr1F	MK738109	AACCTGGGTTACGTCCACTG	224 bp
Mtnr1R		AGCGAACCCACAAAGAGGTT	
Mtnr2F	MK738110	CGTGGTCTTTGTGCTGTTCG	20 bp
Mtnr2R		ATGCGCTTGTACTCGTTCCT	
Mtnr3F	MK738111	CAAGACGATCCTCCTCGCTC	185 bp
Mtnr3R		GGCGAGGCTCCAGATAACAA	
P53F	EU711045.1	GCGGGCTCAGTATTTTGAAGAC	94 bp
P53R		GCTCAGCAGGATGGTCGTCA	
Caspase3F	JQ394697.1	TCGTTCGTCTGTGTCCTGTTGAG	91 bp
Caspase3R		GCTGTGGAGAAGGCGTAGAGG	
BaxF	XM_020094597.1	GCTCCAGAGGATGATAAATAAC	124 bp
BaxR		AAAGTAGAAGAGTGCGACCA	
Bcl2F	XM_020104180.1	TTATCAGCGGCATCTTCATCTC	113 bp
Bcl2R		TTGGCGAGGCGGTGTAATC	
β-actinF	AY008305	CATGTACGTTGCCATCCAAG	138 bp
β-actinR		ACCAGAGGCATACAGGGACA	

Note: Mtnr1: melatonin receptor 1; Mtnr2: melatonin receptor 2; Mtnr3: melatonin receptor 3; Caspase3: cysteinyl aspartate specific proteinase 3; Bcl2: B-cell lymphoma-2; Bax: BCL2-associated X, apoptosis regulator.

## Data Availability

The data are presented in this study and the Appendix A.

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
