# Peer review of "Melatonin Improves Turbot Oocyte Meiotic Maturation and Antioxidant Capacity, Inhibits Apoptosis-Related Genes mRNAs In Vitro"

_antioxidants, 2023, doi:10.3390/antiox12071389_

Round 1

Reviewer 1 Report

The manuscript describes the effect of melatonin exposure in vitro to turbot oocytes. Authors assess several endpoints such as oocyte development, morphology and viability, in combination with the activity and content of antioxydant enzymes and proteins. The main conclusion is that melatonin protect the oocytes and it is proposed as a potential molecule to improve oocyte quality in fish.

The manuscript requires an extensive revision of the English. This is critical aspect authors must address before re-submitting the manuscript. I also have doubts that Antioxidants is the most suitable journal for this work. Indeed, antioxidants activities are just one of the several endpoints studied in the work, but not the main objective of the work. It is Editor´s final decission.

Regarding methodology it is not clear how many samples are analysed for each endpoint. For example, I do not know if for assessing antioxidant activities the sample number is 3 per group or other. This must be clearly indicated for all endpoints. Statistical analysis is based on parametric tests, but if the number of samples per group is 3, the approach followed is completely incorrect. I suggest using non-parametric test when sample size is very small.

Also, it is not mentioned how protein was determined. A mention to the protocol used must be included.

One of the weakest aspects is that authors do not mention the viability of the oocyte cultures. This is crucial in order to understand the results. What was the survival/viability of oocytes in culture conditions?

bax/bcl2 ratio: Why was this ratio calculated? Any reference to that?

Conclusions: Authors in the indicate they were the first one producing a stable oocyte culture in turbots. However, as mentioned above, they do not provide any information about culture viability. What do authors understand about stable curlture conditions?

Legend and section names mistakes: There are several editing mistakes that should be improved. For example:

Table 1: Legend is incorrect. Column 1 (Genes) refers to primer names, no to gene names.

Section 3.2 name: Not correct.

Legend Figure 8: Repeated sentence.

Author Response

 We would like to thank the reviewers for their comments on our work and their suggestions for improving the quality of our MS. We have revised the MS based on their suggestions. The revisions were highlighted in red. The detailed responses to each question raised by reviewer 1 are listed as follows:

Point 1: The manuscript requires an extensive revision of the English. This is critical aspect authors must address before re-submitting the manuscript. I also have doubts that Antioxidants is the most suitable journal for this work. Indeed, antioxidants activities are just one of the several endpoints studied in the work, but not the main objective of the work. It is Editor´s final decission.

Response 1: Good suggestion. We have completed extensive English revisions by editing service. Meanwhile, the revised text was checked by my colleague fluent in English writing. Thank you very much!

Point 2: Regarding methodology it is not clear how many samples are analysed for each endpoint. For example, I do not know if for assessing antioxidant activities the sample number is 3 per group or other. This must be clearly indicated for all endpoints. Statistical analysis is based on parametric tests, but if the number of samples per group is 3, the approach followed is completely incorrect. I suggest using non-parametric test when sample size is very small. Also, it is not mentioned how protein was determined. A mention to the protocol used must be included.

Response 2: Sorry for my incomplete description. Actually, all experiments were repeated three times to confirm the results. Triplicate wells were used for each endpoint or treatment within an experiment. Totally, nine samples per group in the current study. To avoid confusing the readers, the boxplots (Fig 4, 5, 6) be applied to visualize the data distribution and outlier individuals correctly. We also using non-parametric test based on your suggestion, the results showed the similar tendency. The protein was determined by commercial bicinchoninic acid (BCA) detection kits in accordance with the manufacturers' instructions (Nanjing Jiancheng Bioengineering Institute, Nanjing, China). The detailed information was added on lines 143-144, 153-154. Thank you very much!

Point 3: One of the weakest aspects is that authors do not mention the viability of the oocyte cultures. This is crucial in order to understand the results. What was the survival/viability of oocytes in culture conditions?

Response 3: Good question, you are expert. Stable and high viability of turbot oocyte culture in vitro is the basic needing for all experiments in the current study. Fish oocyte culture in vitro is significantly different from mammals due to the species specific and the diversity of fish species. Actually, our preliminary experiment found the dead oocytes and follicles will floating the surface of medium after culture for 3h, the normal oocytes and follicles laid on the bottom the medium during culture in vitro. Meanwhile, the survival rate of turbot oocytes and follicles cultured in vitro more than 95% in the present study. We have added the survival rate of turbot oocytes and follicles culture in vitro treatment with melatonin and DHP in supplemental data (Figure S3) and described on lines 248-250 in revised MS.

Point 4: bax/bcl2 ratio: Why was this ratio calculated? Any reference to that?

Response 4: It has been identified that the alteration of the bax/bcl-2 ratio may affect mitochondrial cytochrome c release and can be predictive of whether or not a cell undergoes apoptosis (Gross et al.,1999; Hildeman et al., 2003). An increase in bax/bcl-2 ratio is a decisive factor that is positively correlated with apoptosis or cell death (Kim et al., 2010; Jin et al., 2011). Thus, we calculate the bax/bcl2 ratio. We have added the reference on lines 391-392 in revised MS.

Reference

Gross, A., McDonnell, J.M., Korsmeyer, S.J., 1999. BCL-2 family members and the mitochondria in apoptosis. Genes Dev. 13, 1899-1911.

Hildeman, D.A., Mitchell, T., Aronow, B., Wojciechowski, S., Kappler, J., Marrack, P., 2003. Control of Bcl-2 expression by reactive oxygen species. Proc. Natl. Acad. Sci.100, 15035-15040.

Kim, H.G., Song, H., Yoon, D.H., Song, B.E., Park, S.M., Sung, G.H., et al., 2010. Cordyceps pruinosa extracts induce apoptosis of HeLa cells by a caspase dependent pathway. J. Ethnopharmacol. 128, 342-351.

Jin, Y., Zheng, S., Pu, Y., Shu, L., Sun, L., Liu, W., et al., 2011. Cypermethrin has the potential to induce hepatic oxidative stress, DNA damage and apoptosis in adult zebrafish (Danio rerio). Chemosphere 82, 398-404.

Point 5: Conclusions: Authors in the indicate they were the first one producing a stable oocyte culture in turbots. However, as mentioned above, they do not provide any information about culture viability. What do authors understand about stable curlture conditions?

Response 5: You are right. Up to now, there is no literatures reported turbot oocyte culture in vitro. We have carries out more preliminary experiments to identify the optimal medium, temperature, pH and other conditional factors (data not shown). The common phenomenon was found, dead oocytes and follicles will be floating the surface of medium within culture for 3h in vitro. The survival rate of turbot oocytes and follicles culture in vitro was over 95% in the present study (Supplemental Figure S3). Just like I have mentioned above fish oocytes and follicles culture in vitro is significantly different from mammals due to the species specific and the diversity of fish species. Meanwhile, oocyte is different from other cells adhering to the wall of the well. Traditional MTT and other viability assay method is not suitable used in the current study. Thus, we focus on the selection of healthy female turbot and oocyte survival rate culture in vitro. To guarantee more scientific description, the “stable” changed into “high survival” on lines 398 in revised MS.

Point 6: Legend and section names mistakes: There are several editing mistakes that should be improved. For example:

Table 1: Legend is incorrect. Column 1 (Genes) refers to primer names, no to gene names.

Section 3.2 name: Not correct.

Legend Figure 8: Repeated sentence.

Response 7: We have carefully revised the legend and section names mistake on lines 180, 239, 292-293 based on your suggestion. Thank you!

All revisions were finished in the revised MS. We thank the reviewer for the expert comments to improve this manuscript.

Reviewer 2 Report

The paper entitled ‘Melatonin improves turbot oocyte meiotic maturation and anti-2 oxidant capacity, inhibits apoptosis-related genes mRNAs in 3 vitro’ shows that melatonin have important potential for improving oocytes maturation and quality. The concept of the role of melatonin in the regulation of fish reproduction, including its impact on oocytes maturation, is not completely novel, because for example it was previously found that melatonin accelerates the action of maturation inducing hormone on oocyte maturation in carp. However, the current study performed on marine fish species – Turbot which production is quite important for European and Asian fish farming market. Therefore, it can be assumed that the presented results may be important from some readers.

However, there are a few points need clarification:

1)      How was the specificity of the PCR amplification verified?

2)      Was the length of the PCR product for Mtnr2 20 bp or rather 204 bp?

3)      How was the reference gene for the analyses obtained? Were any other (than beta actin) reference genes tested?

4)      Were melatonin levels measured in the medium after incubation? Melatonin degrades relatively easily and its level after 24 hours of incubation may be different than at the beginning of the experiment. It is also significant because some of its degradation products may exhibit biological activity.

5)      I believe it should be mentioned in the discussion that there are also other melatonin receptors than those studied in this experiment.

Summarizing, in my opinion, the topic of article generally fit the scope of Antioxidants and the manuscript may be published in after revision made.

Author Response

We would like to thank the reviewers for their comments on our work and their suggestions for improving the quality of our MS. We have revised the MS based on their suggestions. The revisions were highlighted in red. The detailed responses to each question raised by reviewer 2 are listed as follows:

Point 1: How was the specificity of the PCR amplification verified?

Response 1: Good suggestion. We did the melt curve for all used genes, confirming their amplification efficiency prior to the experiment. We have added the amplification plot and melt curve of real time RT-PCR in supplemental data (Figure S4).

Point 2: Was the length of the PCR product for Mtnr2 20 bp or rather 204 bp?

Response 2: Sorry for the lack of special Marker. The length of the PCR product for mtnr2 is 204 bp. We have added the full imags in revised Figure 4. Thank you very much!

Point 3: How was the reference gene for the analyses obtained? Were any other (than beta actin) reference genes tested.

Response 3: Good question. The successful application of qRT-PCR depends on accurate transcript normalization via the selection of suitable reference genes. Six reference genes including, 18S ribosomal RNA (18s), beta-actin (β-actin), elongation factor 1-alpha (ef1α), glyceraldehyde-3-phosphate-dehydrogenase (gapdh), cathepsin D (ctsd), and beta-2-microglobulin (b2m) were selected and expression stability through ovarian development were analyzed via using geNorm, NormFinder and BestKeeper algorithms in our previous study (Gao et al., 2020). Results showed that the best-suited gene combinations for normalization were β-actin and ctsd in the ovary. Meanwhile, use the ctsd as reference genes demonstrated the similar results like β-actin in the current study. Thus, we select β-actin as reference gen in the present study.

Reference

Gao, Y., Gao, Y., Huang, B., Meng, Z., Jia,Y., 2020. Reference gene validation for quantification of gene expression during ovarian development of turbot (Scophthalmus maximus). Sci. Rep. 10: 823.

Point 4: Were melatonin levels measured in the medium after incubation? Melatonin degrades relatively easily and its level after 24 hours of incubation may be different than at the beginning of the experiment. It is also significant because some of its degradation products may exhibit biological activity.

Response 4: Sure, we measure the melatonin levels in the medium after incubation at preliminary experiment. Just like what your said, melatonin degrades relatively easily. The concentration of melatonin in medium remained unchanged during 24 hours incubation, whereas significantly decreased at 30 and 36 hours. That’s one of the reasons that oocyte and follicle were treatment with melatonin for 24 h, not so long times. We have provided these results in supplemental data (Figure S5).

Point 5: I believe it should be mentioned in the discussion that there are also other melatonin receptors than those studied in this experiment.

Response 5: We have mentioned in the discussion section on lines 334-336 based on your suggestion.

All revisions were finished in the revised MS. We thank the reviewer for the expert comments to improve this manuscript.

Reviewer 3 Report

Very high quality MS. I really enjoyed reviewing it. I recommend accepting MS for printing after minor revision. My comments are included in the text. To see them all, please open the file in Acrobat Reader. I have added some suggestions to the Discussion chapter - they may be useful to the authors also in future research and perhaps in developing a system for evaluating oocyte quality and maturity on fish-farms before turbot spawning.

Author Response

We would like to thank the reviewers for their comments on our work and their suggestions for improving the quality of our MS. We have revised the MS based on their suggestions. The revisions were highlighted in red. The detailed responses to each question raised by reviewer 3 are listed as follows:

Point 1: I suggest (but it's just a suggestion) to start the Discussion a bit differently. In my opinion, the knowledge of the biology of oocyte maturation together with the methods of assessing the quality of oocytes in the period of several days before ovulation (especially if they are combined with the synchronization of hormonal stimulation to induce final oocyte maturation [FOM] and ovulation) is an additional value in itself and of great practical aspect. The methods of oocyte maturity assessment used, for example, in cyprinids in the case of turbor will not be useful. But those used for perch fish, if only because of the very similar structure of oocytes - in my opinion, yes.

Response 1: Good suggestion. We have added it on lines 296-298 in revised MS. Thank you very much!

Point 2: Therefore, I suggest the authors consider the possibility of using in the discussion such solutions described for

Eurasian perch, e.g. Reproductive Biology, 2011, 11(3), pp. 194–209; Aquaculture, 2011, 313(1-4), pp. 84-91

pickerch, e.g. Aquaculture Research, 2012, 43(5), pp. 713–721; Reproduction, Fertility and Development, 2012, 24(6), pp. 843-850

ruffe, e.g. Animal Reproduction Science, 2021, 225, 106684

burbot, e.g. Aquaculture, 2022, 548, 737679

And in conclusion - that this research can be used (be helpful) in developing such systems in the future.

Response 2: We have cited these reference in discussion section on lines 303-305. Thank you very much!

All revisions were finished in the revised MS based on reviewer 3 annotation. We thank the reviewer for the expert comments to improve this manuscript.

Reviewer 4 Report

Zhang  et al

 Melatonin improves turbot oocyte meiotic maturation and anti- oxidant capacity, inhibits apoptosis-related genes mRNAs in vitro

Major comments

It is well established that melatonin is a pineal gland neurohormone that direct non-receptor-mediated free radical scavenging activity and is expressed in most organisms. Still it may be of interest to document the effect of melatonin in turbot. The experimentral data presented are solid and of high quality. However, the presentation and discussion of the results need attention before the manuscript may be published. Most of the lingual mistakes and strange wording in the manuscript can be relatively easily amended, but are just too many to be fully specified here. Especially the Discussion, this part contains mistakes in nearly every sentence.

Specific points.

Abstract

Line 11-12;

Write commercial aquaculture OR production during fish farming, not both.

Line 13; Litter, probably should be little

L 14 – 16. Sentence starting; The present study… Don’t include what you want to do, but rephrase to express what you actually achieved. The sentence fits well as ending the Introduction, where a similar but much too detailed sentence could be omitted (L 84 – 89).

L 16, Expressions like “Results showed” should be avoided if possible. Better express actually how the results were achieved. For example use; The cultures showed… or similar.

L 19, plus 4 more times in the Abstract. The word “significantly”, should be used when expressing a statistical fact. Many places it would be more informative if actual numbers are used. Like here, melatonin was in fact 20 times more effective than DHP. This would be evident if molar concentrations for melatonin were used (not g/L). See also line 21.

Materials and methods.

L95; Sexually matured or Sexually mature, there is a distinction that important.

L109; Wording is wrong and the sentence is not complete. Rephrase..

L115; Meanwhile?? Something is missing in this sentences

L116. Ovaries or the ovaries (not The ovary), and were (not was) and cultures (not culture)..

L 119; .. in zebrafish [48,49] and perch [21] with  some modifications.

L 123: … separated in two groups. In one group the follicles were digested, but what was done with the other group?

L 124 – 126, After enough….  Something is missing in the sentence.

L128; … randomly distributed.. is this correct?

L133 … identify the optimal size of denuded oocytes used for in vitro study. Is it more correct to write? .. identify the optimal size of denuded oocytes to be used for in vitro studies.

L134 - 135  Sentence is not complete, and is not well formulated.

L136 Melatonin must also be given in Molar concentration

Results

L198 Write full name at first time for Prevtg to the Mig-nucle.  Same for Atre 199 and Latvtg L200

L198  … values observed… (delete  was)

L 203 204 Sentence; The isolated immature stage… needs attention. Very difficult to understand as it is.

L209 …while more than 80% is 0.5-0.7 mm oocyte. Change to for instance;   of which more than 80% where 0.5-0.7 mm oocytes

L210 Change to; The highest GVBD rate was observed in oocytes

L229- 230 In Fig 1 legend.  Prevtg, previtellogenic; Evtg, 229 early vitellogenic; Latvtg, late vitellogeneic; Mig-nucle, migratory nuclei; Atre, atresia.

L232 Fig 2 legend. ..oocytes..

L233, I immature oocyte, M maturation oocytes. Use same for both – oocytes or oocyte

Maybe change to; removal of follicle cell layers by collagenase treatment.

L242. Give molar concentrations for melatonin. This is more correct and makes it easy to compare with other hormones like DHP. Molar concentrations should be used throughout the text.

L244-245 Melatonin also… The sentence needs attention.

The next sentence, starting However, the GVBD rate did not showed… have several grammatical mistakes.

L291. Fig 6. Rephrase the title. For instance; Expression of Apoptosis-related genes after treatment with melatonin for 24 hours. 

Discussion.

Near every sentence in the Discussionnedds attention. A new, Discussion part must be presented before the manuscript can be accepted.

Conclusions

L399-400  effects of melatonin on  maturation of  oocyte meiotic. Not sure if I understand this sentence. Please rephrase.

L400. Late vitellogenic with diameter oocytes of 0.5-0.7 mm, had a.. Maybe write; Late vitellogenic oocytes with diameter  of…

L412++ The last sentence of the conclusions need attention, as it is very difficult to understand as it is.

Author Response

We would like to thank the reviewers for their comments on our work and their suggestions for improving the quality of our MS. We have revised the MS based on their suggestions. The revisions were highlighted in red. The detailed responses to each question raised by reviewers are listed as follows:

Abstract

Point 1: Line 11-12. Write commercial aquaculture OR production during fish farming, not both. Line 13. Litter, probably should be little

Response 1: We have deleted the word of “production during fish farming” and change “Litter” into “little” in revised MS on lines 12-13. Thank you.

Point 2: Line 14-16. Sentence starting; The present study… Don’t include what you want to do, but rephrase to express what you actually achieved. The sentence fits well as ending the Introduction, where a similar but much too detailed sentence could be omitted (L84–89).

Response 2: Good suggestion. We have rephrased these sentences on lines 13-16, 85-87 in revised MS.

Point 3: Line 16. Expressions like “Results showed” should be avoided if possible. Better express actually how the results were achieved. For example use; The cultures showed… or similar.

Response 3: We have revised it on lines 16 in revised MS. Thank you very much!

Point 4: Line 19. plus 4 more times in the Abstract. The word “significantly”, should be used when expressing a statistical fact. Many places it would be more informative if actual numbers are used. Like here, melatonin was in fact 20 times more effective than DHP. This would be evident if molar concentrations for melatonin were used (not g/L). See also line 21.

Response 4: Good suggestion. We have added it on lines 19, 21 in revised MS.

Materials and methods.

Point 5: Line 95. Sexually matured or Sexually mature, there is a distinction that important.

Response 5: We have corrected the “Sexually matured” into “Sexually mature” on line 92 in revised MS. Thanks.

Point 6: Line 109. Wording is wrong and the sentence is not complete. Rephrase.

Response 6: We have rewritten this sentence on lines 106-107 in revised MS. Thanks.

Point 7: Line115. Meanwhile?? Something is missing in this sentences.

Response 7: Sorry for our mistakes. The word of “Meanwhile” is unnecessary word, we have deleted it.

Point 8: Line 116. Ovaries or the ovaries (not The ovary), and were (not was) and cultures (not culture).

Response 8: Good suggestion. We have revised it on lines 113-114 in revised MS.

Point 9: Line 123. … separated in two groups. In one group the follicles were digested, but what was done with the other group? Line 124-126. After enough….  Something is missing in the sentence.

Response 9: You are expert. The follicles were separated two groups. In one group the follicles were digested by 0.3 mg/ml collagenase (Solarbio, Beijing China) in a shaking water bath at 20 °C for 40 min to remove follicular cell layer, the other group did not undergo collagenase treatment and remained in in fresh Ca2+-Mg2+-free DMEM medium. We have added related information on lines 120-125 in revised MS.

Point 10: Line 128. … randomly distributed.is this correct?

Response 10: Randomly distributed means the same size of oocytes or follicles were randomly distributed in wells of a 24-well plate. To avoid confusing readers, we have rephrased it on lines 127-130.

Point 11: Line 133. … identify the optimal size of denuded oocytes used for in vitro study. Is it more correct to write? . identify the optimal size of denuded oocytes to be used for in vitro studies.

Response 11: Thank you for your suggestion. We have revised it on line 133.

Point 12: Line 134-135. Sentence is not complete, and is not well formulated.

Response 12: We have rewritten it on lines 134-135 in revised MS. Thanks.

Point 13: Line136. Melatonin must also be given in Molar concentration

Response 13: We have revised it on line 137 in revised MS.

Results

Point 14: Line 198. Write full name at first time for Prevtg to the Mig-nucle. Same for Atre 199 and Latvtg L200.

Response 14: Good suggestion. We have revised it on lines 198-202. Thank you very much!

Point 15: Line 198. … values observed… (delete was)

Response 15: We have deleted it. Thank you very much!

Point 16: Line 203-204 Sentence; The isolated immature stage… needs attention. Very difficult to understand as it is.

Response 16: Good suggestion. We have rewritten it on lines 204-205 in revised MS.

Point 17: Line 209. …while more than 80% is 0.5-0.7 mm oocyte. Change to for instance; of which more than 80% where 0.5-0.7 mm oocytes.

Response 17: We have revised it on lines 210-211 based on your suggestion. Thank you very much!

Point 18: Line 210. Change to; The highest GVBD rate was observed in oocytes

Response 18: We have changed it on line 211 based on your suggestion. Thanks!

Point 19: Line 229-230. In Fig 1 legend.  Prevtg, previtellogenic; Evtg, 229 early vitellogenic; Latvtg, late vitellogeneic; Mig-nucle, migratory nuclei; Atre, atresia.

Response 19: We have revised it on lines 230-231 in revised MS. Thank you very much!

Point 20: Line 232. Fig 2 legend...oocytes. Line 233. I immature oocyte, M maturation oocytes. Use same for both – oocytes or oocyte Maybe change to; removal of follicle cell layers by collagenase treatment.

Response 20: We have revised it on lines 233-235. Thank you very much!

Point 21: Line 242. Give molar concentrations for melatonin. This is more correct and makes it easy to compare with other hormones like DHP. Molar concentrations should be used throughout the text.

Response 21: Good suggestion. The unit of melatonin was changed into molar concentrations in revised MS on lines 243-244, 247, 275. Meanwhile, the Fig3, Fig5, Fig6 were also changed it in revised text.

Point 22: Line 244-245. Melatonin also… The sentence needs attention. The next sentence, starting However, the GVBD rate did not showed… have several grammatical mistakes.

Response 22: Sure,you are right. We have revised it on lines 246-247 based on your suggestion. Thank you very much!

Point 23: Line 291. Fig 6. Rephrase the title. For instance; Expression of Apoptosis-related genes after treatment with melatonin for 24 hours.

Response 23: We have rephrased the title on lines 293-294 in revised text.

Discussion.

Point 24: Near every sentence in the Discussion needs attention. A new, Discussion part must be presented before the manuscript can be accepted.

Response 24: I am very sorry for our roughly English writing in discussion section. We have completed extensive English revisions by editing service. Meanwhile, the revised MS was checked by our colleague fluent in English writing.

Conclusions

Point 25:L399-400…effects of melatonin on maturation of oocyte meiotic. Not sure if I understand this sentence. Please rephrase. L400. Late vitellogenic with diameter oocytes of 0.5-0.7 mm, had a. Maybe write; Late vitellogenic oocytes with diameter of… L412++ The last sentence of the conclusions need attention, as it is very difficult to understand as it is.

Response 25Good suggestion. We have revised it on lines 402-403, 415-416.

All revisions were finished in the revised MS. We thank the reviewer for the expert comments to improve this manuscript.

Round 2

Reviewer 4 Report

The authors have updated the manuscript as suggested, and it appears now to be suitable for publication after a few corrections.

Specific comments  

L71; Maybe change to; Turbot Scophthalmus maximus is an economically important flatfish species

L215 follicles that are 0.5-0.7 mm…  follicles that were…  OR follicles of 0.5-0.7 mm…

L217  … oocytes with 0.5-0.7 mm in diameter were used for..  Remove “with” or use; oocytes with diameter of 0.5-0.7 mm were used for

L247-8  Check grammar, several mistakes

L 249 – use The survival rate

L250 … among control, DHP and melatonin treatment group..  Write for instance; …for the  control, DHP and the melatonin treatment group…

L357 Excessive  not  Excessively

L375++  Apoptosis paragraph needs attention, I’m far from an apoptosis expert and find the text confusing. There are several mistakes and inaccurate sentences.  

L377 Sentence needs attention.  .. are important apoptosis-related molecules  OR .. are important molecules in the apoptotic signaling pathway.

L379  not factors Bax, use singular. For Bcl-2 it is more complicated, both singular and plural could be correct  as Bcl-2 is a factor, but also names a family of factors (including Bax). Maybe write  …Bcl2 factors  (not factors Bcl2). Please specify.

L381  maybe write  … members of the Bcl2  family.  Check if Bcl-2 or Bcl2 is correct.

L387 and L392 .. and bax/bcl2 in sheep. Maybe write bax/bcl2 ratio 

L396 -399 Last sentence need attention.

Conclusions need some attention. Especially grammatical time varies in the text. Best use passed term as in the three first sentences, also in the following sentences.

For example; L408  suppressed…   L409. activated.. Use passed term as above.
L411 promoted L412 inhibited

Author Response

  We would like to thank the reviewers for their comments on our work and their suggestions for improving the quality of our MS. We have revised the MS based on their suggestions. The revisions were highlighted in red. The detailed responses to each question raised by reviewers are listed as follows:

Point 1: Line 71. Maybe change to; Turbot Scophthalmus maximus is an economically important flatfish species; Line 215. follicles that are 0.5-0.7 mm…  follicles that were…  OR follicles of 0.5-0.7 mm… Line 217. … oocytes with 0.5-0.7 mm in diameter were used for.  Remove “with” or use; oocytes with diameter of 0.5-0.7 mm were used for.

Response 1: Good suggestion. We have revised it on lines 72, 222-223 in revised MS. Thank you very much!

Point 2: Line 247-8. Check grammar, several mistakes

Response 2: We have rephrased these sentences on lines 255-256 in revised MS. Thank you very much!

Point 3: Line 249. use the survival rate.

Response 3: We have corrected the “rates” into “rate” on line 257 in revised MS. Thanks.

Point 4: Line 250. … among control, DHP and melatonin treatment group...  Write for instance; …for the control, DHP and the melatonin treatment group…

Response 4: We have changed it on line 258 based on your suggestion. Thanks!

Point 5: Line 357. Excessive not Excessively

Response 5: We have changed it on line 367 in revised MS. Thanks. 

Point 6: Line L375++. Apoptosis paragraph needs attention, I’m far from an apoptosis expert and find the text confusing. There are several mistakes and inaccurate sentences.

Response 6: We have revised it on lines 385-386 in revised MS.

Point 7: Line 377. Sentence needs attention.  ... are important apoptosis-related molecules OR ... are important molecules in the apoptotic signaling pathway.

Response 7: We have revised it on line 387-388 based on your suggestion. Thanks! 

Point 8: Line 379. not factors Bax, use singular. For Bcl-2 it is more complicated, both singular and plural could be correct as Bcl-2 is a factor, but also names a family of factors (including Bax). Maybe write …Bcl2 factors (not factors Bcl2). Please specify. Line 381. maybe write … members of the Bcl2 family.  Check if Bcl-2 or Bcl2 is correct.

Response 8: We have rewritten it on lines 387, 389, 392 in Revised MS. Thanks.

Point 9: Line 387 and 392. … and bax/bcl2 in sheep. Maybe write bax/bcl2 ratio

Response 9: We have changed it on line 398 in revised MS. Thank you very much!

Point 10: Line 396-399. Last sentence need attention. Conclusions need some attention. Especially grammatical time varies in the text. Best use passed term as in the three first sentences, also in the following sentences. For example; Line 408. suppressed…   Line 409. activated... Use passed term as above. Line 411. Promoted. Line 412. Inhibited.

Response 10: Thank you for your suggestion. We have revised it on lines 407-409, 420, 423 in revised MS.

All revisions were finished in the revised MS. We thank the reviewer for the expert comments to improve this manuscript.
